# Plant-Root Exudate Analogues Influence Activity of the 1-Aminocyclopropane-1-Carboxylate (ACC) Deaminase Gene in *Pseudomonas hormoni* G20-18^T^

**DOI:** 10.3390/microorganisms11102504

**Published:** 2023-10-06

**Authors:** Ajay Madhusudan Sorty, Fani Ntana, Martin Hansen, Peter Stougaard

**Affiliations:** Department of Environmental Science, Aarhus University, 4000 Roskilde, Denmark; fnt@bactolife.com (F.N.); martin.hansen@envs.au.dk (M.H.)

**Keywords:** ACC deaminase, regulation, wheat root exudates, promoter fusion, *Pseudomonas*, metabolomics

## Abstract

Plants exposed to abiotic stress such as drought and salinity produce 1-aminocyclopropane-1-carboxylic acid (ACC) that is converted into the stress hormone ethylene. However, plant growth-promoting bacteria (PGPB), which synthesize the enzyme ACC deaminase, may lower the ACC concentration thereby reducing the concentration of ethylene and alleviating the abiotic stress. The PGPB *Pseudomonas hormoni* G20-18^T^ (previously named *P. fluorescens* G20-18) harbors the genes *acdR* and *acdS* that encode regulation and synthesis of ACC deaminase, respectively. Regulation of the *acdS* gene has been investigated in several studies, but so far, it has been an open question whether plants can regulate microbial synthesis of ACC deaminase. In this study, small molecules in wheat root exudates were identified using untargeted metabolomics, and compounds belonging to amino acids, organic acids, and sugars were selected for evaluation of their influence on the expression of the *acdS* and *acdR* genes in *P. hormoni* G20-18^T^. *acdS* and *acdR* promoters were fused to the fluorescence reporter gene mCherry enabling the study of *acdS* and *acdR* promoter activity. In planta studies in wheat seedlings indicated an induced expression of *acdS* in association with the roots. Exudate molecules such as aspartate, alanine, arginine, and fumarate as well as glucose, fructose, and mannitol actively induced the *acdS* promoter, whereas the plant hormone indole-3-acetic acid (IAA) inhibited expression. Here, we present a model for how stimulatory and inhibitory root exudate molecules influence *acdS* promoter activity in *P. hormoni* G20-18^T^.

## 1. Introduction

Ethylene acts as an important plant growth hormone that plays a vital role in breaking seed dormancy [1]. However, constant high levels of ethylene can also induce several detrimental effects with a predominant negative impact on root elongation and early senescence in plants [2,3]. Plant tissues use S-adenosyl methionine (SAM) to produce 1-aminocyclopropane-1-carboxylic acid (ACC) that serves as a precursor in the biosynthesis of ethylene. In addition to synthesis in developing seedlings, ACC is also increasingly formed in plants under stress. Plant-associated bacteria exhibit a unique capability to cleave this precursor molecule and thus to regulate the subsequent ethylene formation. This reaction is regulated by the enzyme ACC deaminase [4]. Therefore, being related to plant sustenance under stressful circumstances is considered one of the key determinants of microbial plant growth-promoting (PGP) characteristics and also a key factor regulating the microbe’s associative lifestyle, for instance in Rhizobia [5]. The presence of the ACC deaminase gene, *acdS*, has been documented in a variety of endo- and epiphytic bacteria including members of the *Pseudomonas* group [6,7]. ACC deamination appears to vary among microbial species. For example, in root nodules, *acdS* expression only accounted for 2–10% of the activity expressed by free-living microbes [8]. However, *acdS* function in symbionts is mainly focused on enhancing and maintaining nodulation [9] and may not necessarily participate in stress mitigation through lowering ethylene levels. Thus, although the levels of activity differ significantly with habitat and associative interactions with the host, the expression of ACC deamination among PGPR appears relatively evenly distributed.

In most organisms, *acdS* is located close to its regulator gene, *acdR*, that encodes a leucine-responsive regulator (Lrp), and this *acdR*–*acdS* gene cluster has been mapped to the chromosome in most Beta- and Gammaproteobacteria, whereas the cluster is located on plasmids in Alphaproteobacteria [10,11]. In silico analyses of the DNA region between the *acdR* and *acdS* genes in *Pseudomonas putida* UW4, *Azospirillum lipoferum* 4B, and *Methylobacterium radiotolerans* JCM 2831 showed the presence of putative binding sites for Lrp, cAMP receptor protein (CRP), and fumarate-nitrate reduction regulator (FNR) [8,12,13,14], and in vitro binding experiments showed that AcdR/Lrp indeed bound to the region between *acdR* and *acdS* [8,14]. In vivo experiments confirmed that the *P. fluorescens* UW4 *acdS* gene, when expressed recombinantly in *Escherichia coli*, was regulated by *E. coli* AcdR/Lrp, CRP, and FNR [12,13] and results obtained by Pringent-Combaret et al. [11] indicated that expression of *A. lipoferum* 4B *acdS* might be regulated by AcdR/Lrp and FNR but not CRP. The expression of *acdS* is highly dependent on oxygen availability, substate concentration, feedback from the reaction products, and also catabolite repression effects due to specific substrates [15]. The regulatory mechanisms of *acdS* are well-explained in the model organism *P. putida* UW4 by Li et al. [15].

However, from the plants’ perspective, both the expression as well as the regulatory mechanisms of *acdS* need further exploration in order to understand the highly intricate plant–microbial crosstalk and plant-regulatory influence on microbial gene expression, physiology, and metabolism. Since it is very likely that exudates from plant roots are involved in the regulation of the *acdS* and/or *acdR* genes, we analyzed root exudate molecules from wheat and investigated if some of the exudate molecules were involved in regulating *acdS* and/or *acdR* expression using promoter fusion technology with the fluorescence reporter mCherry.

## 2. Materials and Methods

### 2.1. Bacterial Strains and Growth Media

The *P. hormoni* G20-18^T^ strain (previously named *P. fluorescens* G20-18; now described as a novel species, *P. hormoni* G20-18^T^, manuscript in press) used throughout this study was cultivated at 20–25 °C in Lysogenic Broth (LB) or minimal medium (M9). *E. coli* HST08 (Stellar^TM^, Takara Bio Europe) was used in the construction of the *acdS* replacement mutant and the promoter fusion constructs. *E. coli* cells were cultivated in LB at 37 °C.

### 2.2. Construction of Promoter Fusions

Promoter fusion strains of *P. hormoni* G20-18^T^ were constructed to investigate the characteristic influence of plant metabolites on *acdS* expression in detail. Mining the annotated genome sequence of strain G20-18 (GenBank accession no. CP075566) showed the presence of an *acdS* gene (locus tag KJF94_09105) and an *acdR* gene (locus tag KJF94_09100) (Appendix A). The 200-base pair fragment comprising the *acdS* and *acdR* promoters was cloned into the reporter plasmid pSEVA237R [16] using In-Fusion cloning (Takara Bio Europe) with primers 5′-GCGGCCGCGCGAATTGTGGCTTCTGCACAATAAAAATATG-3′ and 5′-CGACTCTAGAGGATCGACTCTGCTCCTTGTTATTGG-3′ (Figure 1). Gene replacement in which the *P. hormoni* G20-18^T^ *acdS* gene was replaced by the gentamycin resistance gene (Δ*acsS*) was carried out as described by Hennessy et al. [17] and Michelsen et al. [18] using the gene replacement vector pEX100T. The promoter fusion constructs and the *acdS* mutant, Δ*acsS*, were sequenced prior to conducting expression experiments. The activity of the *acdR* and *acdS* promoters in *P. hormoni* G20-18^T^ was assayed by measuring fluorescence from the mCherry reporter gene using a fluorescence plate reader (CLARIOstar Plus–BMG Labtech, Ortenberg, Germany) as reported by Hennessy et al. [17].

### 2.3. Production and Characterization of Wheat Root Exudates

Wheat root exudates were produced in hydroponic as well as in sterile soil conditions. The hydroponic growth was achieved by growing surface sterilized seeds (4% sodium hypochlorite for 8 min followed by 70% ethanol for 30 s) in sterile mass-spectrometry-grade water for 10 days. The water was changed every second day, and the final fraction at the 10th day was collected and enriched using solid-phase extraction as mentioned below and subjected to untargeted metabolomic analysis by high-resolution mass spectrometry.

The root exudate production in sterile soil was performed by growing surface sterilized seeds for 4 weeks under axenic conditions, followed by gently harvesting the roots. Excess soil on the root surface was removed by gentle shaking until a fine layer of ≤3 mm remained on the root surface. Although a soil cylinder with a maximum size of 4.0 mm around the roots is considered the rhizosphere [19], we considered a soil cylinder of ≤3 mm for the collection of the rhizosphere as the experimental plants were grown in pots in this study. The roots, along with the adhering rhizosphere soil layer were then immediately flash frozen in liquid nitrogen, transferred to −80 °C overnight, and freeze-dried at −100 °C under vacuum for 72 h. The rhizosphere soil from the dried roots was then collected by gentle tapping on the container. A one hundred-milligram fraction of the soil was then weighed and used for the extraction of root exudates. The solvent system for the extraction of root exudates was specifically standardized for polar to moderately polar compounds, to achieve optimal extraction of mobile root exudate moieties. Briefly, the solvent system for extraction contained 0.05% aqueous formic acid (A); 50% methanol in A (B); and 95% methanol in A (C). Each soil sample was sequentially extracted using each of the three extraction solvents at a proportion of 1:2 (*w/v*). The contents were vortex mixed at 3000 rpm for 15 min, followed by bath-ultrasonication for another 15 min, and the supernatants were collected. Finally, the solvent phase was isolated using high-speed centrifugation at 20,000× *g* for 20 min; the three aliquots were pooled and subjected to solid-phase extraction using 1cc HLB-30 mg extraction cartridges (Oasis, Waters–Ireland) and eluted with 1 mL methanol. The elutes were evaporated to dryness under nitrogen flow at room temperature, and the resultant pellet was dissolved in 1 mL 5% methanol. The extracts were then filtered using a Ø13 mm 0.22 μm PVDF syringe filter (Millex-Durapore, Merck, Kenilworth, NJ, USA). Ten microliters of each sample was injected into a UHPLC-Orbitrap-HRMS/MS platform (ThermoFisher Scientific, Waltham, MA, USA). The UHPLC (Ultimate3000, ThermoFisher Scientific) was fitted with a Kinetex biphenyl analytical column (2.1 × 100 mm, 2.6 µm, Phenomenex, Denmark) and operated with a biphasic acetonitrile/water gradient at a flow rate of 400 µL/minute. The mass spectrometer was operated in positive and negative electrospray ionization mode with data-dependent acquisition (Top3) mode with stepped collision (30 and 70 NCE). The data were processed using in-house Compound Discoverer and GNPS pipelines as described elsewhere [20]. The initial peak picking from the raw data into centroided form was performed using MSConvert (URL (1 August 2023), https://proteowizard.sourceforge.io/download.html). Features with annotated (confidence level 2) [21] molecular identities were pin picked for analyzing their influence on the expression of either *acdR* or *acdS* promoters. Similarly, the knowledge on root exudates from the available literature was also considered during the selection of test compounds that could influence the expression of *acdR* and *acdS* promoters.

### 2.4. Activity of acdR and acdS Promoters under the Influence of Root Exudate Analogues

The influence of synthetic root exudate analogues on the promoter expression was achieved using a growth system constructed in a 96-well microtiter plate design. The growth substrate contained 1x M9 medium with or without ammonium or glucose, (20% *v/v* stock mix of 0.25 M Na_2_HPO_4_, 0.11 M KH_2_PO_4_, and 0.0425 M NaCl; 2% *v/v* 1 M MgSO_4_; 0.01% *v/v* 1 M CaCl_2_; 10 or 25 mM test compounds; with or without 0.1% NH_4_Cl and 100 ppm of ACC). Furthermore, in order to determine the influence of the test compounds alone, phosphate-buffered saline (PBS) or aqueous systems were also used. Cells in the logarithmic growth phase were prepared by growing overnight in LB medium, pelleting and washing with sterile milliQ water, and resuspending in milliQ water in such a way that the final concentration of the inoculum in the test reaction medium was adjusted to approximately 0.05–0.07 OD_600_. Expression from the *acdR* and *acdS* promoters was then monitored in terms of rise in mCherry fluorescence (excitation at 570–15 nm; and emission at 620–20 nm), while cell growth was measured in terms of OD_600_. The measurements were recorded for 12–48 h at 3.75- or 30-min scan-cycles in a fluorescence plate reader (CLARIOstar Plus—BMG Labtech, Ortenberg, Germany) at 25 °C. The data were analyzed in terms of relative growth vs. fluorescence, and a ratio of fluorescence/growth dynamics.

### 2.5. Wheat Seedling in Planta Studies

In planta studies in wheat seedlings were carried out to determine the associative behavior of the strain *P. hormoni* G20-18^T^, and to study the dynamic expression of the *acdS* promoter during the associative lifestyle. Wheat seeds were surface sterilized as described above and subsequently treated with bacterial suspensions (~10^6^ CFU − OD_600_ = 0.1) for 60 min. A Tn7 mCherry-tagged wild type *P. hormoni* G20-18^T^ was provided by Prof. Thomas Roitsch (Dept. of Plant and Environmental Sciences, University of Copenhagen, Copenhagen, Denmark). The seeds were air-dried on sterile filter paper and cultivated in Petri dishes (*n* = 10 × 3) containing sterile filter paper beds. Seeds were allowed to germinate at 20 °C in the dark for 5 days, followed by light/dark exposure for 16/8 h for the next 5 days. The early roots were harvested from the cotyledon and 100 mg of fresh roots was washed with 5 mL of PBS containing 0.01% Tween-20 at 180 rpm for 30 min. The solution containing epiphytic cells was preserved and the roots were further washed with sterile milliQ water 5 times. The roots were then crushed in 5 mL of sterile milliQ water, the debris was allowed to settle, and the cleared solutions were read in a fluorescence reader as mentioned above.

### 2.6. Statistical Analysis

All the experiments were conducted in triplicate unless specified. ANOVA was applied using SPSS 16.0 (Windows 8.0, URL www.spss.com, (accessed on 1 August 2023)). The differences at the 95% confidence levels were considered significant.

## 3. Results and Discussion

The untargeted metabolomic data analysis of root exudates showed the presence of a variety of organic compounds belonging to different metabolic groups, viz., primary and secondary metabolites. The chemistry of the identified biomolecules predominantly highlighted the presence of important classes including low-molecular-weight organic acids, phenolics, amino acids, benzoxazinoides, plant hormones, etc. (cf. Table 1 and Appendix A). Many of the observed metabolites, e.g., amino acids and plant hormones, are already known for their diversified functioning within the rhizosphere microzone.

In addition to the identified root exudate molecules in this study, we also included the root-secreted biomolecules already reported in the literature or the parent molecules (e.g., sugars) that were observed in modified forms in the root exudate composition [25,26,27] as well as their role in influencing *acdS* expression [28]. These molecules included amino acids and low-molecular-weight organic acids that mainly comprise primary metabolites and thus may also have a significant influence on the *acdS* and *acdR* promoters. Furthermore, known *acdS* regulatory compounds, such as ACC itself, α-ketobutyrate, leucine, and ${\mathrm{NH}}_{4}^{+}$, were also included. 

The analysis of the genome sequence of *P. hormoni* G20-18^T^ revealed that the ACC deaminase gene, *acdS*, was co-localized with the Lrp-like regulator gene, *acdR* (Appendix A). This gene organization has been reported before for a number of Proteobacteria, including *Pseudomonas* spp. [11]. The region between the *acdR* and *acdS* genes contains the promoters for the *acdR* and *acdS* genes including putative binding sites for regulatory molecules. This region was amplified by PCR and inserted into the reporter plasmid pSEVA237R [16]. The fragment was inserted in both orientations; in one orientation, the *acdR* promoter transcribed the mCherry gene, and in the opposite orientation, the mCherry gene was transcribed from the *acdS* promoter. The mCherry promoter fusion plasmids were transformed into wild-type *P. hormoni* G20-18^T^ (*acdS^+^*) cells and in the replacement mutant, Δ*acdS*. However, since wild-type cells with active ACC deaminase reduce the concentration of ACC with the concomitant production of ${\mathrm{NH}}_{4}^{+}$ and α-ketobutyrate, experiments with wild-type cells and ACC would complicate the picture when long incubations were conducted (data not shown); thus, we used Δ*acdS* for the subsequent experiments. Both *acdR* and *acdS* promoter fusions were included in the study. However, as the *acdR* promoter was shown to be expressed constitutively, whereas the *acdS* promoter was regulated by a number of molecules, only the results from the *acdS* promoter fusion are shown below. 

### 3.1. Influence of Nitrogenous Compounds on acdS Promoter Activity

Addition of ACC alone in M9 medium (no C or N supplements) confirmed earlier reports, which showed that ACC induced the *acdS* promoter (Appendix A) [4,13,29,30]. The end products of the ACC deaminase reaction, α-ketobutyrate and ammonium, also influenced *acdS* promoter activity. Figure 2A shows that the induction of the *acdS* promoter by 100 ppm ACC was similar in experiments with or without the addition of 0.1% ${\mathrm{NH}}_{4}^{+}$ or ${\mathrm{NO}}_{3}^{-}$. When 25 mM α-ketobutyrate and 100 ppm ACC were added, the promoter activity increased by a factor of 3–4, but this stimulation was lowered to approximately half if ${\mathrm{NH}}_{4}^{+}$ (or ${\mathrm{NO}}_{3}^{-}$) was added to the reaction (Figure 2B). This interaction between ammonium and α-ketobutyrate was also observed when glucose was added or if nitrate was substituted for ammonium (Figure 2C,D). This phenomenon underscores the interaction of ammonium nitrogen in regulating *acdS* activity, and also links the presence of *acdS* sequences to the *Nif* regulatory regions in symbiotic bacteria [2]. However, increased expression due to amino acids and ACC over that with the ACC alone without any nitrogen source (Figure 3) points towards the probable influence of the nitrogen component from the amino acids. 

### 3.2. Influence of Amino Acids on acdS Promoter Activity

A significant amount of carbon is involved in amino acid fluxes in wheat. For instance, Phillips et al. [22] described fluxes of 16 amino acids from wheat and other plants under axenic and microbially associated conditions. The authors reported methionine as the lowest (4 nmol g^−1^ root h^−1^) and alanine as the highest efflux molecule (60 nmol g^−1^ root h^−1^). We also noticed the presence of amino acids in root exudates in a qualitative form (Table 1). Thus, the abundance of amino acids in root exudates generated an interest in evaluating their influence on *acdS* promoter activity. The expression of *acdS* dramatically changed following the addition of amino acids (Figure 3). As anticipated, no expression was observed in the absence of ACC. Previous publications have pointed to leucine as an inhibitor [7,11,12,13]. However, here, we show that the addition of 25 mM leucine and 100 ppm ACC to growing cells increased the *acdS* promoter activity by 78%, when compared to cells only induced with ACC. A similar, small stimulation was also observed by Honma [28], who reported that ACC deaminase activity in cells without leucine was 0.22 × 10^−3^ units/mg but if leucine was supplemented, the activity increased to 0.26 × 10^−3^ units/mg. Furthermore, Figure 3 shows that the addition of either fumarate, alanine, valine, arginine, or isoleucine further increased the *acdS* promoter activity. Honma [28] also reported that the specific activity of ACC deaminase in cells increased upon addition of valine. The conflicting results here, when compared to the literature, could be due to different concentrations of leucine used [11,13] and the fact that some of the previous results were based on in vitro gel retardation experiments [7]. However, the action of Lrp regulators may be very complex. Recent reports [31,32,33] describe how leucine may act as an inducer or inhibitor. In a recent review on Lrp, Ziegler and Freddolino describe that leucine acts on *E. coli* Lrp by causing a shift from a hexadecamer (16 mer) to an octamer (8 mer) if leucine is added [31]. Furthermore, they conclude that with respect to DNA binding, the general consensus in the field is that leucine increases the cooperative binding of Lrp to DNA, but overall, it reduces the affinity of Lrp to DNA. Finally, they point out that all in vitro assays on Lrp required anywhere from 1 to 10 mM leucine to elicit an effect on Lrp, whereas the typical concentration of leucine in cells is on the order of 0.1 mM. Furthermore, other amino acids, such as aspartate and arginine as demonstrated earlier in wheat root exudates by Phillips et al. [22], also stimulate *acdS* promoter activity (Figure 3). Prigent-Combaret et al. also reported that the addition of arginine increased the *acdS* mRNA levels in *A. lipoferum* 4B when measured by semiquantitative RT-PCR [11]. 

l-amino acid isomers without ACC did not induce the *acdS* promoter, aligning with the available knowledge [34,35]. However, the characteristic response of *acdS* observed in this study in the presence of ACC plus l-amino acids indicates a yet-unknown mechanism, especially synergistic, or antagonistic (probably dependent on interactions of structural confirmations of amino acids –, e.g., branched chain, hydrophobic groups, etc.) that could either up or down regulate expression (Figure 3B). Further, fumarate, alanine, and aspartic acid induced expression here (Figure 3A,B); both alanine and aspartic acid are intermediates in the same biosynthetic branch, where fumarate acts as a precursor [36]. Thus, fumarate, aspartate, and alanine together indicate a clear positive influence on the cellular biochemistry of *acdS*. Additionally, the response in the presence of fumarate also underscores the involvement of FNR activity in *acdS* regulation. The phenomenon also aligns with the higher *acdS* expression seen in endophytic environments (Figures 6 and 7). As FNR proteins are major contributors to the oxygen response, particularly for switching from an aerobic to anaerobic mode of metabolism [37], their involvement in adaptation to the endophytic lifestyle and *acdS* regulation highlights that the *acdS* response in an endophytic habitat is tightly regulated and may not be solely dependent on the ACC abundance in cellular environments.

### 3.3. Influence of Sugars on acdS Promoter Activity

Following the evaluation of amino acids, optimization of the system to investigate the influence of individual sugar moieties was carried out. Although we could not detect sugars in native form within the wheat root exudates, evidence from the available literature indicates a dominance of sugars in root-secreted carbon in plants such as maize [38]. Therefore, to generate a wider context for microbial *acdS* interactions in general, we also included representative sugars in this study. A combination of glucose, ammonium, nitrate, and α-ketobutyrate was tested in presence of 100 ppm ACC. Glucose without N sources but with ACC induced the activity of the *acdS* promoter, while inclusion of either of the N sources lowered the expression (Figure 2C,D). The addition of α-ketobutyrate indeed slightly enhanced the expression, indicating that the presence of nitrogen has an influence on the *acdS* promoter. Sugars other than glucose influenced *acdS* promoter activity. Figure 4 shows that the other sugars could also positively induce the *acdS* promoter. Within the first 11–12 h (Figure 4A), glucose induced the promoter the most, but after 24 h of incubation, fructose was the dominant inducer. The observation that fructose is a better inducer when compared to glucose has also been made by [11] in *A. lipoferum* 4B. Mannitol was almost as good an inducer as glucose, whereas l-rhamnose only displayed a small stimulation compared to the no sugar addition. Sucrose, however, in the present study, showed no induction although it was shown earlier to be present in the root exudates [38].

### 3.4. Influence of Indole-3-Acetic Acid (IAA) on acdS Promoter Activity

The major goal of this study is to study *acdS* expression from the plants’ perspective and thus, the measurements typically focus on biomolecules originating from the plant root exudations. Therefore, considering the rhizosphere microzone as a general habitat of the microbe, we used very low concentrations of the test molecules (10–25 mM) over the reported concentrations (e.g., 1% as reported by Honma [28]. Other molecules that microorganisms may encounter in the rhizosphere are plant hormones such as IAA. Here, we show that 1500 ppm IAA inhibited *acdS* promoter activity (Figure 5).

### 3.5. acdS Promoter Activity in Epiphytic vs. Endophytic Cells

Another important aspect to be considered in the *acdS* expression dynamics is linking the current results to the lifestyle of *P. hormoni* G20-18^T^. This bacterium was isolated from an arctic grass [39], and although there is significant knowledge available regarding beneficial interactions of *P. hormoni* G20-18^T^ in crop plants [40,41], its associative behavior in wheat is not known. Here, we show that *P. hormoni* G20-18^T^ can adopt both epiphytic and endophytic lifestyles. Figure 6A shows that *P. hormoni* G20-18^T^ tagged with Tn7 mCherry could be retrieved both from the root surface and from the endophytic compartment. Similarly, fluorescent mCherry reporter proteins were observed in the epiphytic and endophytic compartments, respectively, when *acdS* promoter fusion cells were inoculated on wheat roots. The mCherry fluorescence from the Tn7-tagged constructs and from *acdS* promoter fusions was higher in epiphytic extracts compared to endophytic extracts, indicating that more cells were attached to the root surface than present in the endophytic compartment. However, as the ratio of *acdS* promoter fluorescence/Tn7-tagged fluorescence was higher in endophytic extracts than in epiphytic extracts, this could indicate that the *acdS* promoter was more active inside the plant root cells than on the outside (Figure 6B). This observation is in line with previous reports that showed higher *acdS* activity in cells cultivated in oxygen-limited conditions [11,12]. Furthermore, the results also align with the observations reported by [5] during *Sinorhizobium meliloti* colonization in the root zone of *Medicago sativa*. However, unlike Rhizobia, the range of activity of *acdS* in the case of the *P. hormoni* G20-18^T^ might not be limited to establishment and maintenance of association, whereas the induced expression indicates active channelizing of ACC even from the epiphytic zone that could balance the internal ACC levels in the roots [8]. This phenomenon provides important links to the previous report mentioning that *P. hormoni* G20-18^T^ mediated drought tolerance in tomato [41]. The results therefore align with the model of microbe-mediated deamination of ACC in plants that was proposed by Glick et al. [42], who mentioned that seeds and/or roots leak significant quantities of ACC that is utilized by rhizosphere microbes.

### 3.6. Model for Regulation of acdS Promoter Activity in P. hormoni G20-18^T^

The results presented here led us to propose the following hypothesis for the interactions between *P. hormoni* G20-18^T^ and wheat roots (Figure 7). In the root under normal growth conditions without abiotic stress, Figure 7A, there exists an intricate balance between ethylene and IAA biosynthesis and subsequent localization of IAA to the elongation zone. During stress (Figure 7B), induction of ethylene leads to ethylene-responsive cascades that induce auxin transport proteins and induce localization of IAA within the root elongation zone, causing ethylene-induced inhibition of root elongation. However, in stressed plant roots growing in the presence of *P. hormoni* G20-18^T^ (Figure 7C), ACC produced by the plant will be channeled into the ACC deaminase-producing bacterial cells where ACC is converted to ammonia and α-ketobutyrate. Furthermore, root exudate molecules such as sugars (e.g., fructose, glucose, and mannitol) and amino acids (e.g., aspartate, alanine, and arginine) may further stimulate this conversion of ACC, and in the endophytic compartment, fumarate may further induce synthesis of ACC deaminase. Thus, the lower ACC concentration will result in lower ethylene-induced inhibition of root elongation. However, this will only work if ACC deaminase-producing bacteria are present on or in the root before the onset of the abiotic stress. If the bacteria associate with stressed plant roots, it is likely that the high IAA concentrations might induce an inhibitory influence on the production of ACC deaminase and no bacterium-induced lowering of the ethylene concentration will occur (Figure 7D). Overall, the plant IAA could exhibit a higher influence due to its ethylene-dependent specific localization in the roots that determines root growth and development. Although a general concentration of IAA ranges on the order of 250 pM g^−1^ of wheat roots on a fresh weight basis [43], the localization could be highly site-specific along the roots and might lead to the development of IAA-rich zones. Moreover, under a natural scenario, an association of high-IAA-producing microbes could further augment the local IAA concentration at the site of root colonization; thus, the phenomenon depicted in Figure 7C may occur along the root regions with IAA hyperactivity. Further, within the rhizosphere and endophytic habitat, *acdS* expression occurs in the presence of a complex mixture of biomolecules that could influence *acdS* activity in either a positive or negative manner. Therefore, net *acdS* activity under such a highly complex chemical scenario still remains unknown. 

The influence of IAA in the strains with both the IAA^+^ and *acdS^+^* traits remains an open question. In the case of IAA^+^, *acdS^+^* microbial strains, the interactions might involve even more intricate molecular cascades. However, microbial IAA, being a secondary metabolite [44], could exhibit higher concentrations during the late growth phases extending from the late-log to stationary stage. Meanwhile, *acdS* expression seems more dependent on the induction by ACC, and could continue to be expressed even during the log phase of growth. Furthermore, as evident from the fumarate responses (Figure 3A), during the adaptation to an endophytic lifestyle, the process could probably trigger low oxygen-induced metabolic rearrangements through FNR proteins that also seem involved in *acdS* regulation. Therefore, the results strongly endorse the need to thoroughly investigate the *acdS* trait in *P. hormoni* G20-18^T^, and microbial *acdS* trait in general in the context of rhizosphere and/or endophytic microhabitats where both the inducers and inhibitors are found together in a cocktail of metabolites.

## 4. Conclusions

The results show that the *acdS* promoter activity in *P. hormoni* G20-18^T^ is dynamic and can be regulated through a range of commonly occurring biomolecules, particularly those belonging to the sugars, amino acids, and phenolics including plant hormones. However, in all cases, ACC is required as a primary inducer of *acdS*; thereafter, the expression is further modulated in the presence of different biomolecules. Principally, the associative behavior and metabolite chemistry exert a significant influence on the *acdS* promoter, and hence, can be regarded as key determinants of microbial *acdS* traits under natural conditions. Furthermore, a decline in expression in the presence of the reaction product—particularly ammonia—indicates the need for detailed investigations in an agricultural scenario where N inputs are frequent and can cause a higher abundance of ammonia nitrogen. The aligning trends of *acdS* in the presence of ammonia and nitrate also highlight the possible involvement of cellular nitrogen fluxes in *acdS* regulation. The results also underscore the need to investigate the interactions in planta under drought conditions.

## Figures and Tables

**Figure 1 microorganisms-11-02504-f001:**
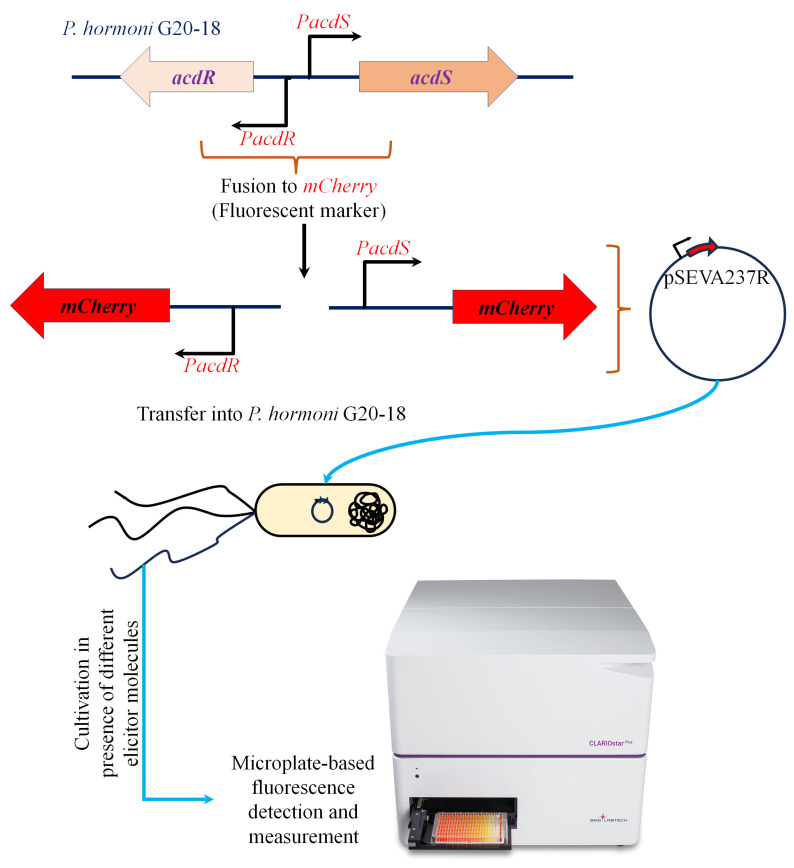
Schematic representation of the workflow adapted to generate an mCherry-tagged *acdS* and *acdR* promoter fusion in *P. hormoni* G20-18^T^.

**Figure 2 microorganisms-11-02504-f002:**
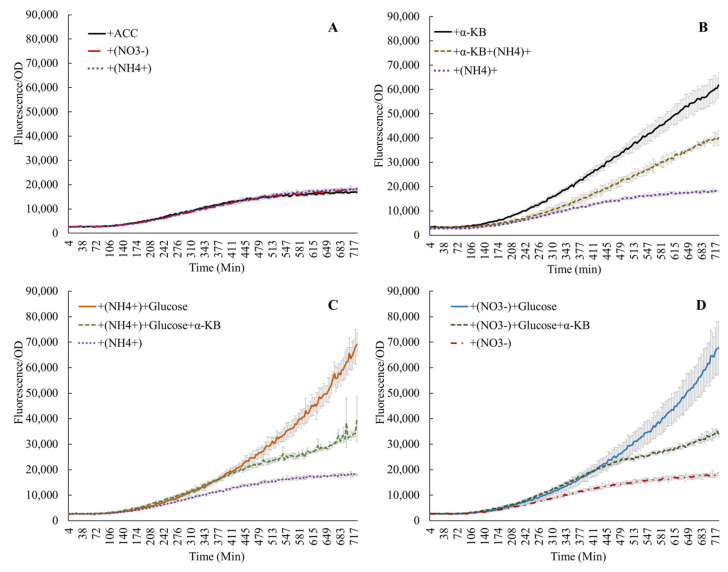
Activity of the *acdS* promoter in the presence of nitrate, ammonium, or ACC (**A**); α-ketobutyrate (α-KB) lowers the expression only in the presence of nitrogen (**B**), while the induced expression due to glucose is lowered in the presence of both the reaction products of ACC deaminase (**C**,**D**).

**Figure 3 microorganisms-11-02504-f003:**
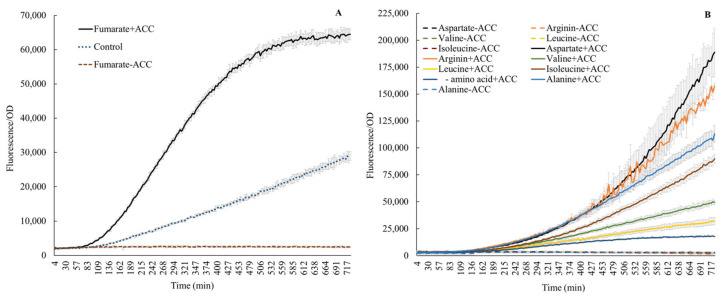
Effect on *acdS* promoter activity by fumarate (**A**) and amino acids (**B**), with or without ACC. Control (**A**): without fumarate, but with ACC.

**Figure 4 microorganisms-11-02504-f004:**
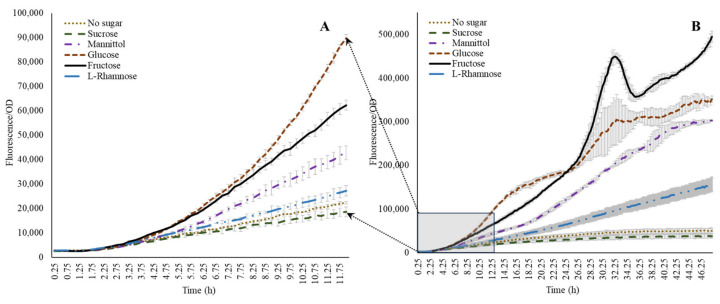
Effect of sugars on *acdS* promoter expression. Note that glucose dominates during initial growth (**A**), while the induction is taken over by fructose during the later growth stages (**B**).

**Figure 5 microorganisms-11-02504-f005:**
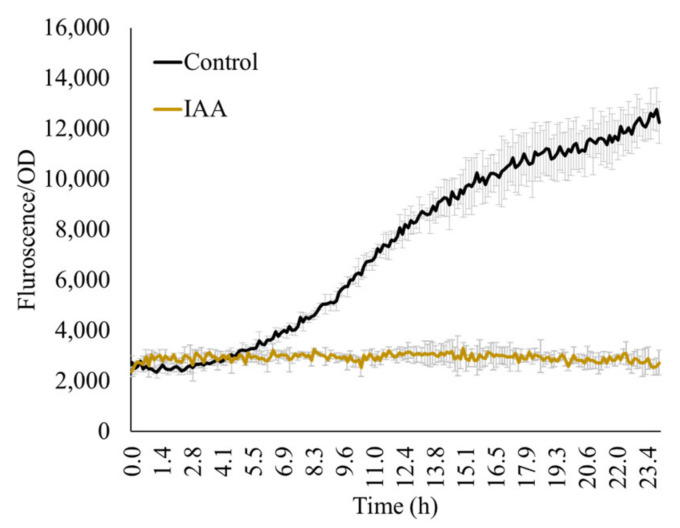
IAA (1500 ppm) in the presence of ACC (100 ppm) inhibits expression from the *acdS* promoter. Control: 100 ppm ACC without IAA.

**Figure 6 microorganisms-11-02504-f006:**
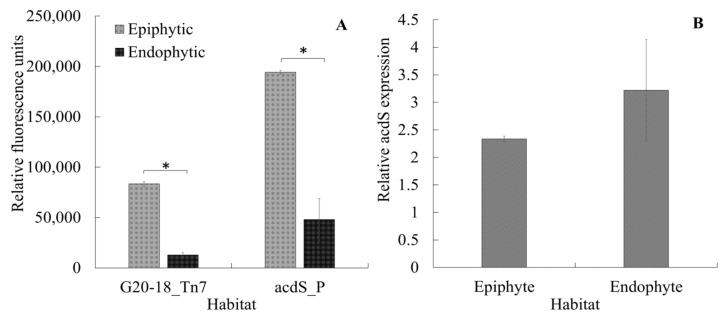
Colonization of the mCherry-tagged *acdS^+^ P. hormoni* G20-18^T^ strain on and in the wheat seedling roots (A-G20-18_Tn7). *acdS* promoter expression in promoter fusion strain (**A**) and relative expression of the *acdS* promoter (*acdS* promotor fluorescence/Tn7 tagged fluorescence) in epi- and endophytic environments (**B**). * Indicates that the differences are statistically significant at *p* ≤ 0.05.

**Figure 7 microorganisms-11-02504-f007:**
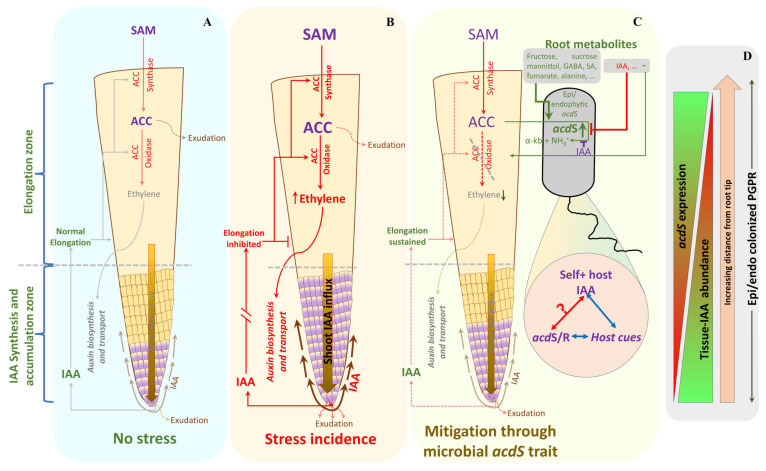
A model of microbial acdS interactions with the host plant under abiotic stress conditions. Although IAA has an inhibitory effect on the *acdS* promoter, the regulation through self-IAA originating from the microbial cells still needs to be investigated. (SAM: S’-adenosyl methionine).

**Table 1 microorganisms-11-02504-t001:** An overview of the biomolecules that could influence *acdS* expression.

Molecule	Formula	*m*/*z* *	Reference
Aspartic acid ^#^	C_4_H_7_NO_4_	134.05	This study
Valine ^#^	C_5_H_11_NO_2_	118.08	This study
Arginine ^#^	C_6_H_14_N_4_O_2_	174.10	This study
Isoleucine ^#^	C_6_H_13_NO_2_	132.10	This study
Alanine ^#^	C_3_H_7_NO_2_	90.05	[22]
Leucine ^#^	C_6_H_13_NO_2_	132.10	This study
Tryptophan	C_11_H_12_N_2_O_2_	205.09	This study
Phenylalanine	C_9_H_11_NO_2_	166.08	This study
Glucose ^#^	C_6_H_12_O_6_	179.05	This study
Fructose ^#^	C_6_H_12_O_6_	179.05	This study
Sucrose ^#^	C_12_H_22_O_11_	341.10	This study
Mannitol ^#^	C_6_H_14_O_6_	221.04	This study
L-Rhamnose ^#^	C_6_H_12_O_5_	164.06	[23]
α-Ketobutyric acid ^#^	C_4_H6O_3_	102.09	[24]
Fumaric acid ^#^	C_4_H_4_O_4_	116.35	This study
GABA	C_4_H_9_NO_2_	104.07	This study
Malic acid	C_4_H_6_O_5_	133.01	This study
Indole-3-acetic acid ^#^	C_10_H_9_NO_2_	175.06	This study
Salicylic acid	C_7_H_6_O_3_	137.02	This study
Succinic acid	C_4_H_6_O_4_	101.02	This study
Azelaic acid	C_9_H_16_O_4_	189.11	This study
*Trans*-zeatin	C_10_H_13_N_5_O	218.1	This study
DIMBOA	C_9_H_9_NO_5_	210.04	This study
DIBOA	C_8_H_7_NO_4_	182.04	This study
Jasmonic acid	C_12_H_18_O_3_	211.13	This study

^#^ Molecules evaluated for their influence on *acdS* promoter expression in *Pseudomonas hormoni* G20-18^T^. * Standard *m*/*z* values.

## Data Availability

The genome sequence of *P. hormoni* G20-18^T^ (*P. fluorescens* G20-18) had been determined before GenBank accession no. CP075566. The *acdS* gene (locus tag KJF94_09105) and *acdR* gene (locus tag KJF94_09100) sequences are derived from the genome sequence.

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
