# Peer review of "Plant-Root Exudate Analogues Influence Activity of the 1-Aminocyclopropane-1-Carboxylate (ACC) Deaminase Gene in Pseudomonas hormoni G20-18T"

_microorganisms, 2023, doi:10.3390/microorganisms11102504_

Round 1

Reviewer 1 Report

The study title:

 Plant-root exudate analogues influence activity of the 1-amino-2 cyclopropane-1-carboxylate (ACC) deaminase gene in Pseudo-3 monas fluorescens G20-18"

 In this study, small molecules wheat root exudates were identified using untargeted metabolomics, and compounds belonging to amino acids, organic acids, and sugars were selected for evaluation of their influence on the expression of the acdS and acdR genes in P.  fluorescens G20-18. acdS and acdR promotors were fused to the fluorescence reporter gene mCherry enabling the study of acdS and acdR promotor activity.

I find the research interesting and well structured, I recommend minor modifications as below:

 In line 140 delete the word  "done "

In " The initial peak picking from the raw data into centroided form was performed  done using

In material and methods section

What is the main objective of " Wheat seedlings in planta studies"  ?

I recommend writing the main objective of each section in material and methods to help the reader follow the experiment clearly

In Figure 7  : write the full meaning of SAM " what does it mean"?

In general figures, resolution should be enhanced

Best of luck 

Author Response

In line 140 delete the word "done" in "The initial peak picking from the raw data into centroided form was performed done using”

-Line 140 (now line 152) revised as suggested

In material and methods section

What is the main objective of "Wheat seedlings in planta studies"? I recommend writing the main objective of each section in material and methods to help the reader follow the experiment clearly

-The objectives are included in the methods section. e.g. line 88-89; 161-162; 179-180 in the revised submission.

In Figure 7: write the full meaning of SAM " what does it mean"?

-the abbreviation is explained in the Figure 7 caption.

In general figures, resolution should be enhanced

-we have revised the figures for higher quality.

Reviewer 2 Report

The work of Sorty and coauthors entitled ¨ Plant-root exudate analogues influence activity of the 1-aminocyclopropane-1-carboxylate (ACC) deaminase gene in Pseudomonas fluorescens G20-18¨ did very good work to elucidate the expression, induced by root exudates or analogs, of the acdS and acdR genes by fusion of promoters with a fluorescent marker. The construction was carried out in a plasmid that is later transferred to the G20-18 strain of Pseduomonas fluorescens.

The article in general is very well structured, very solid and the results are very clear. I believe that there are a few works that I have reviewed of this type of plant growth-promoting bacteria of the genus Pseudomonas. These results are very valuable due to the lack of information on the gene expression of the gene that codes for the ACC deaminase (acdS) enzyme and its regulator, important in the modulation of ethylene and environmental stress in plants. The last figure of their model is very didactic (congratulations).

I only have a few minor comments:

1. The abstract can be improved by describing more the results.

2.     The authors mention that they did an analysis of the root exudates using untargeted metabolomic analysis by high-resolution mass spectrometry. These results should be presented in a table in the main body of the text. I couldn't see the complementary data during the review of the manuscript, maybe they have it somewhere. But it would be important to see them firsthand. In the table, perhaps those analogs used in the expression analysis could be added or marked in bold.

3.     Please correct NH4+ (valency is not subscript, but a small number). The same for others (e.g. NO3-).

4.     Figure 6. Please correct acdS gene in italics.

5.     Figure 6. Please describe the statistical analysis.

6.     Figure 7. Please correct acdS gene in italics in legend.

Author Response

The abstract can be improved by describing more the results.

-we have revised the abstract with inclusion of additional results

The authors mention that they did an analysis of the root exudates using untargeted metabolomic analysis by high-resolution mass spectrometry. These results should be presented in a table in the main body of the text. I couldn't see the complementary data during the review of the manuscript, maybe they have it somewhere. But it would be important to see them firsthand. In the table, perhaps those analogs used in the expression analysis could be added or marked in bold.

-the table S1 is now included in the main text in the revised manuscript.

Please correct NH4+ (valency is not subscript, but a small number). The same for others (e.g. NO3-).

-The formulae are now revised for the correct formatting using the formula function throughout the text

Figure 6. Please correct acdS gene in italics.

-the formatting has been corrected in the revised manuscript

Figure 6. Please describe the statistical analysis.

-the statistics part is now included in the methods section in the revised manuscript. Please find the revised lines 196-199

Figure 7. Please correct acdS gene in italics in legend.

-Formatting of the figure 7 legend has been revised

Reviewer 3 Report

The authors of this manuscript present interesting research regarding plant-root exudate analogues influence activity of the 1-amino-2 cyclopropane-1-carboxylate (ACC) deaminase gene in Pseudomonas fluorescens G20-18. Introduction and material and methods sections are well described. Results presented in tables and figures are clear and quite explicable. I believe that the authors discuss and explain the findings of their work compering their work with other similar works in a well written discussion. However, according to my opinion, discussion section could be separated form results. Thus, the text needs very few revisions. I think that this research study could add further interest to the researchers worldwide.

Title

Please check title format. I think that every first letter of each word must be capitalized. Please also justify the text, some paragraphs are not justified with others.

Abstract

COMMENT:

The abstract describes well the purpose and the scope of this research work.

Introduction

Introduction section is well written and, in my opinion, give the appropriate information.

Please modify paragraph’s style according to author’s instructions. Apply to the rest of the text.

Line 32       [2,3].      please use comma and apply to the rest of the text

Line 56       [8,12-14], please delete gap and apply to the rest of the text

Line 95       Hennessy et 95 al., [17].

Line 98       I think that Figures font size should be 9 and not 10…Please check author’s instructions and apply to the rest of the text

Material and Methods

Results and Discussion

Please check author’s guidance. I think that results and discussion should be separated sections

Discussion

I think that discussion should be a separated section. Discussion section is well organized. The authors compare their findings with other similar research work.

Conclusions

References

COMMENT:

Please check reference list once again in order to be sure that is according to author’s instruction.

Author Response

Please modify paragraph’s style according to author’s instructions. Apply to the rest of the text.

  • Has been modified

Line 32 [2,3]. please use comma and apply to the rest of the text

  • Has been changed

Line 56 [8,12-14],please delete gap and apply to the rest of the text

  • Has been changed

Line 95 Hennessy et 95 al., [17].

  • Has been changed

Line 98 I think that Figures font size should be 9 and not 10. Please check author’s instructions and apply to the rest of the text

  • Font size has been changed to 9

Results and Discussion

Please check author’s guidance. I think that results and discussion should be separated sections

  • We agree that separate Results and Discussion sections normally are preferred. However, we believe that it is easier for the reader when results and discussion are presented in the same section. The methods used are very similar (same promotor fusion, same mCherry fluorescence output) but the treatments/additions are very different in the individual sections. If the Results and Discussion of the results were separated, we believe that it would be difficult for the reader to remember the individual results when reading the Discussion. Therefore, we would like to keep the combined Results and Discussion section.

Discussion

I think that discussion should be a separated section. Discussion section is well organized. The authors compare their findings with other similar research work.

  • See above.